# Expression Profiles of Hepatic Immune Response Genes in HEV Infection

**DOI:** 10.3390/pathogens12030392

**Published:** 2023-03-01

**Authors:** Yasmin Badshah, Maria Shabbir, Khushbukhat Khan, Hashaam Akhtar

**Affiliations:** 1Department of Healthcare Biotechnology, Atta-ur-Rahman School of Applied Biosciences, National University of Sciences and Technology, Islamabad 44000, Pakistan; 2Global Health Security Agenda (GHSA), National Institutes of Health (NIH), Islamabad 44000, Pakistan

**Keywords:** hepatitis E virus, immune response genes, viral infection

## Abstract

Hepatitis E is a liver inflammation caused by infection with the hepatitis E virus (HEV). Every year, there are an estimated 20 million HEV infections worldwide, leading to an estimated 3.3 million symptomatic cases of hepatitis E. HEV viral load has been studied about the disease progression; however, hepatic the host gene expression against HEV infection remains unknown. **Methods:** We identified the expression profiles of hepatic immune response genes in HEV infections. Fresh blood samples were collected from all the study subjects (130 patients and 124 controls) in 3ml EDTA vacutainers. HEV viral load was determined by a real-time PCR. The total RNA was isolated from the blood using the TRIZOL method. The expression of theCCL2, CCL5, CXCL10, CXCL16, TNF, IFNGR1, and SAMSN1 genes was studied in the blood of 130 HEV patients and 124 controls using a real-time PCR. **Results:** Gene expression profiles indicate high levels of CCL2, CCL5, CXCL10, CXCL16, TNF, IFNGR1, and SAMSN1 genes that might lead to the recruitment of leukocytes and infected cell apoptosis. **Conclusion:** Our study demonstrated distinct differences in the expression profiles of host immune response-related genes of HEV infections and provided valuable insight into the potential impact of these genes on disease progression.

## 1. Introduction

The hepatitis virus destroys liver tissue due to inflammation which may be acute or chronic [1]. Hepatitis comes in five different types, i.e., A, B, C, D, and E. Some of the infections are spread by sexual contact, which is the leading cause of death in most patients, where patients harboring the hepatitis virus infection develop liver failure [2]. The hepatitis E virus is a positive, single-stranded, and non-enveloped RNA virus with a 7.2kb genome that is entirely transmitted [3] through ingested contaminated water.

An estimated 939 million people worldwide have been infected with the hepatitis E virus (HEV), and between 15 and 110 million people now have a current or active case of the infection [3]. Globally, nearly 70,000 people die annually due toHEV [4]. Outbreaks of HEV are more prevalent in underdeveloped regions than in developed ones. Their incidence vary greatly by race, income level, diet, access to clean water and sanitation, and place of origin [5]. Exposure to hepatitis E was reported in 14–26% of an otherwise healthy pediatric group [6]. HEV reports related to prevalence vary in different studies. Nonetheless, HEV was shown to cause acute hepatitis in 20–22% of adults and 2.4% of children [7].

Some clinical manifestations of HEV are hepatomegaly, pruritis, malaise, and fever. These symptoms are further confirmed by laboratory findings based on increased alkaline phosphatase activity and bilirubin levels [8].

The most common genotypes of HEV are HEV-1 to HEV-4, which are human pathogens and have various geographical distributions [9]. An HEV infection triggers the immunological response that includes adaptive and innate lymphoid cells and innate lymphoid cells-1. These all work together against HEV [10]. Previous investigations indicated that hepatitis infections influence the expression of hepatic host genes which may favor the establishment and the progression of the disease [11]. The relationship between HEV infections and host hepatic genes has not been established. Therefore, the present study aims to investigate the potential influence of HEV infections on the expression of host immune response genes (CCL2, CCL5, CXCL10, CXCL16, TNF, IFNGR1, and SAMSN1). The outcomes of the study facilitated us in gaining valuable insight into the mechanism of HEV infections, which, after a further detailed investigation, could help in determining a biological marker for the disease prognosis and therapeutics.

## 2. Materials and Methods

### 2.1. Sample Collection

The current research was carried out at the Atta-ur-Rahman School of Applied Biosciences (ASAB) at the National University of Science and Technology (NUST) in Rawalpindi, Pakistan, in conjunction with the Capital Development Authority (CDA) Hospital in Islamabad, Pakistan, the Railways Hospital (IIMCT) in Rawalpindi, Pakistan, and the Holy Family Hospital in Rawalpindi, Pakistan. The Institutional Review Board (IRB), ASAB, approved the project.

The peripheral blood of 130 HEV-infected patients and 124 healthy individuals were collected in EDTA vacutainer tubes (BD Vacutainer^®^ San Jose, CA, USA) and immediately stored at −20 °C. Patients with co-morbidity were excluded, and samples confirmed by physicians to be HEV-infected were solely obtained. Furthermore, patients with HEV genotype 1a were included in the study. The rest of the genotypes were excluded.

### 2.2. RNA Isolation and Realtime PCR

Total RNA isolation from whole blood was performed using the TriZol reagent, and the whole procedure was conducted on ice to avoid degradation. Afterward, 1 µL of the sample was loaded after blanking the device with nuclease-free water to determine the concentration and purity of the extracted RNA using NanoDrop 2000 (Thermo Fisher Scientific, Waltham, MA, USA). Samples with an A260/A280 ratio between 1.8 and 2.0 depicted pure RNA and were used for cDNA synthesis.

For cDNA synthesis, a 20 µL reaction mixture microcentrifuge tube was comprised of 1 µL OligodT20 {(Random Hexamer, 1 µL dNTP mix (2.5 mM}, and 1000 ng (1 µL) extracted RNA, and the reaction was proceeded for 5 min at 65 °C in a thermocycler, followed by ice incubation. In the next step, 2 µL of the 10X reaction buffer, 1 µL of DTT (100 Mm), 0.5 µL of RNase inhibitor, and 1 µL of RTasewas added to the microcentrifuge tube and volume makeup through the water provided in the kit. It was further processed in the thermocycler for 50 min at 42 °C and then for 10 min at 70 °C. The obtained cDNA was stored at −20 °C.

The expression analysis of CCL2, CCL5, CXCL10, CXCL16, TNF, IFNGR1, and SAMSN1 was performed using a real-time PCR. The primers used for the expression analysis were as follows: 

CCL2 forward 5′-CACCTGCTGCTACTCATTCACTGG-3′, reverse 5′-CTTCTTTGGGACACCTGCTGCTG-3′, CCL5 forward 5′-GACACCACTCCCTGCTGCTTTG-3′, reverse 5′-CTCTGGGTTGGCACACACTTGG-3′,CXCL10forward 5′-CCAAGTGCTGCTGTCGTTCTCTG-3′,reverse 5′-GGTCTCAGCGTCTGTTCATGGAAG-3′, CXCL16 forward 5′-CAGTTTCAGAGCACCCAGCAGTC-3′, reverse 5′-GCCTAGCCTCCAGACCATAGCC-3′, TNF forward 5′-CCGAGGCAGTCAGATCATCTT-3′, reverse 5′-AGCTGCCCCTCAGCTTGA-3′,IFNGR1 forward 5′-TGTAGCGGATAATGGAAC TCTTTT-3′,reverse 5′-AATTTGGCTCTGCATTAT T-3′,SAMSN1 forward, 5′-TCCCTCAAAGCCAGTGACTC-3, and reverse 5′-GCCACAGAATGGTCCTGAAT-3′,

A 20 µL reaction mixture was prepared using WizPure qPCR master mix (SYBR), and it comprised 10 µL of SYBR, 0.4 µL of forward and reverse primers, 2 µL of cDNA, and 6.8 µL of nuclease-free water for a total volume makeup up to 20 µL. The thermocycler conditions were set as follows: initial denaturation at 95 °C for 10min, followed by 40 repeated cycles of denaturation at 95 °C for 30 s, annealing at 60 °C for 30 s, and extension at 72 °C for 30 s. All samples were run in triplicate.

### 2.3. Viral RNA Extraction and qRT-PCR

FAVOREGEN KIT^®^ was used for extracting the viral RNA from the blood and analyzing the viral load in patient samples. An amount of 500 µL of VEN buffer was added into 200 µL of blood already taken in an Eppendorf tube and vortexed for a few seconds. The next vortex was followed by adding 500 µL of 75% ethanol in the tube, and then the sample was transferred to the spin column and centrifuged at 8000 rpm for a minute. The steps of adding wash buffer 1 and 2, followed by centrifugation, were repeated twice. Then, 50 µL of RNase-free water was added and centrifuged at 8000 rpm for 1 min. The RNA in the sample was transferred from filter to tube and stored at −20 °C.

The viral RNA was quantified and amplified simultaneously using qRT-PCR, and it was reverse transcribed into cDNA through the RTase enzyme. SYBR green was used as a dye, and the fluorescence intensity corresponded to the sample quantity. With time, the fluorescence changed and was recorded against the cycles. The amplification progress was depicted in the plot. For the sample preparation, 6 µL of the sample, 12 µL of reverse transcriptase enzyme, and 43 µL of Master mix were added into a PCR tube. Two standards of known concentration (one lower and the other upper limit) were used to determine the viral load in a sample. After the comparison of the viral concentration with the standard curve, a linear graph was obtained with copy number of the template molecules (x-axis) plotted against the cycle threshold (CT) (y-axis). The normalization of the expression was achieved through the housekeeping gene GAPDH.

### 2.4. Statistical Analysis

Statistical analysis was performed via GraphPad Prism version 8.0.1 and Microsoft Excel 2016 (Microsoft cooperation, Redmond, WA, USA). The expression of genes in patients as well as healthy individuals was calculated through the double-delta method, and the fold change was determined. A two-way ANOVA was applied to determine the significance. The mean ± standard deviation (SD) along with the ANOVA was performed to find the association of the mean viral load in different categories, such as age and gender, and to determine the significant association with age or gender. To perform the analysis, patients were distributed into high and low viral load categories, and each group was then evaluated for its association with the viral load.

Similarly, a simple t-test was performed to find the association of gene expression with viral load. Patients were distributed into low and high gene expression groups and evaluated for their significant association with the viral load. Patients with a viral load above the mean viral load were distributed in increased viral load groups. In contrast, patients with a viral load below the mean viral load were distributed into low viral load groups. Ct values above 25 were taken as expression down-regulation, and Ct values below 25 were accepted as expression up-regulation. All the graphs were plotted through GraphPad.

Liver function markers (ALT, ALP, and Bilirubin) were compared between healthy individuals and HEV patients through GraphPad and a simple-t-test was employed to determine the association. Similarly, a principal component analysis was performed through RStudio using *Procomp* command and the relationship between the immune response genes’ expression and liver dysfunction in HEV was determined.

### 2.5. Pathway Construction

Based on the expression results, a cellular pathway was proposed indicating the gene expression’s influence on the cell’s functioning. The pathway was constructed through InkScape [12], and data regarding the pathway nodes and branches were obtained through KEGG [13,14].

## 3. Results

### 3.1. Participants’ Information and Mean Viral Load among Patients

The peripheral blood of 130 HEV patients and 124 healthy controls was taken. Among them, 74 patients were females with a mean age of 37 ± 13 years, and 56 were males with a mean age of 36 ± 11 years with a significance of <0.0001. The mean viral load calculated was mean ± SD 31.6 × 10^7^ ± 11.3 × 10^8^ copies/mL (Table 1).

Based on the obtained viral load, patients below the mean viral load divided into the low viral load group, and patients with a viral load above the mean were distributed into the high viral load group. Most patients had viral loads below the mean, with a minimum of 2.1 × 10^4^ IU/mL and a maximum of 74.3 × 10^8^ copies/mL (Figure 1a). The difference in the means between the two groups was significant (*p* = 0.0014). The viral load of HEV was evaluated for its association with patient gender and age. The analysis indicated the significant association of HEV viral load with females (*p* = 0.00) and ages 20–50 (*p* = 0.005). The viral load concerning gender and age are shown in Figure 1b,c.

### 3.2. Association of Host Immune Response Genes with HEV

The relation of host immune response genes (CCL2, CCL5, CXCL10, CXCL16, TNF, IFNGR1, and SAMSN1) with HEV viral load was investigated. The expression analysis of these genes in patients and healthy individuals indicated that all genes were down-regulated in the healthy individuals, and their expression was up-regulated in patients (Figure 2 and Table 2).

The gene expression below the Ct value 25 was considered down-regulated while above 25 was up-regulated. The mean viral load in patients with genes’ up-regulation and down-regulation were compared and analyzed. The outcomes depicted that the mean viral load was higher in patients having up-regulation of the genes CCL2 (mean viral load 33.6 × 10^7^, *p* = 0.008), CCL5 (mean viral load 35.3 × 10^7^, *p* = 0.005), CXCL10 (mean viral load 35.9 × 10^7^, *p* = 0.006), CXCL16 (mean viral load 46.3 × 10^7^, *p* = 0.002), and IFNGR1 (mean viral load 36.1 × 10^7^, *p* = 0.003). The inverse relationship was found between viral load and gene expression of TNF and SAMSN1. The down-regulation of TNF and SAMSN1 was associated with a higher mean viral load of 42.4 × 10^7^ (*p* = 0.03) and 32.9 × 10^7^ (*p* = 0.18), respectively. The association of immune response genes and HEV viral load is depicted in Figure 3.

### 3.3. Host Immune Response Genes Expression as per Patient Age and Gender

The expression of all host-immune genes (CCL2, CCL5, CXCL10, CXCL15, TNF, IFNGR1, and SAMSN1) was found to be up-regulated in patients in comparison to the host. However, in comparison to males, the expression of these genes was higher among HEV-infected female patients in both the low and high viral load groups (Figure 4a). Similarly, among different age groups, the expression of all genes was significantly up-regulated in patients with a low viral load and age between 20 and 50 (*p* > 0.0001) (Figure 4b).

### 3.4. Relation between Liver Function Markers and Immune Response Genes in HEV Patients

The level of liver function markers (ALT, ALP, and bilirubin) was also studied in HEV patients’ blood. In comparison to the healthy controls, the levels of all markers were higher in HEV patients which indicated the abnormality in liver functioning (Figure 5). In the present study, the relationship between the genes’ expression and normal functioning of the liver was evaluated through principal component analysis (PCA) which is shown in Figure 6 and Figure 7. The principal component analysis graph normally has two dimensions, one for each main component. The x-axis stands for the first dimension that is immune response genes, while the y-axis represents the second dimension; that is, liver function markers. Each data point’s location in the plot reflects its scores on the two components, allowing the correlations between the underlying variables to be deduced. The analysis indicated that all variables belonged in the first two dimensions where immune response genes had a negative relation with liver function markers. However, the genes SAMSN1, IFNGR1, CCL5, and TNF contributed the most in promoting liver function abnormality.

### 3.5. Pathway

The up-regulation of the immune response genes (CCL2, CCL5, CXCL10, CXCL15, TNF, IFNGR1, and SAMSN1) was observed in response to the HEV infection. The influence of the up-regulation of genes was further assessed through pathway analysis. The outcome hinted towards in itiatinga pathway that leads to the recruitment of leukocytes which help the body to fight the HEV infection. This pathway was initiated through TNF signaling that, by using ERK and NF-κβ pathway, leads to the expression of CCLs and CXCLs, causing the recruitment of leukocytes. Furthermore, the infected cells also promote the expression of interferon-gamma (IFNG) which activates the apoptotic pathways via the INFG-receptor, leading to cell apoptosis. Hence, the up-regulation of genes prepares the cell to fight the HEV infection and helps to mitigate the infection by initiating immune responses (Figure 8).

## 4. Discussion

The most prevalent cause of acute viral hepatitis globally is the hepatitis E virus (HEV) infection, which may induce both acute and chronic hepatitis. It can also induce extrahepatic symptoms such as renal and neurological illness [15]. In the United States, its high prevalence is reported among the elderly suffering from chronic liver disease (CLD) [16]. Numerous studies have highlighted the influence of hepatitis C and B on the immune responses in infected individuals, allowing for a better understanding of the disease [17,18,19]. However, there is no literature that has indicated the HEV infection effect on the expression of host immune response genes such as CCL2, CCL5, CXCL10, CXCL15, TNF, IFNGR1, and SAMSN1. Therefore, in the present study, we aimedto investigate the association between HEV infection and the expression of host hepatic immune response genes such as CCL2, CCL5, CXCL10, CXCL15, TNF, IFNGR1, and SAMSN1.

The blood sample of 130 HEV patients was obtained after their written and oral consent was obtained, and the viral load of the HEV was determined. The mean viral load obtained was 31.6 × 10^7^ copies/mL. Previously, in the United States and Canadian populations, the HEV viral load was reported to be between the range of 2 and 30.8 × 10^2^ IU/mL [20]. The viral load data can be helpful in determining the limit of detection for the blood transfusion. In Germany, the limit of detection set to 21 × 10^2^ IU/mL facilitated reducing the blood transfusion-mediated transfer of the HEV infection [21]. Further investigation of the obtained data in the present study could help devise the detection strategy of HEV at a lower viral load.

The impact of patients’ physical characteristics, such as age and gender, on viral load, was also studied. The outcome highlighted the high viral load in females and among patients between 20 and 50 years of age [22]. However, in the German population, males are reported to be at higher risk of HEV infection. Likewise, German patients between the ages of 41 and 51 were more at risk of disease [22]. In another study, older age and consumption of undercooked or uncooked meat are highlighted as a risk factor for an HEV infection [23]. Because pregnancy causes changes in a woman’s immune system, pregnant women have an increased chance of contracting HEV, which may lead to a more severe version of the illness. This risk is owing to the fact that pregnant women are more likely to have a higher viral load [24]. Due to their decreased capacity to fend against infections, those whose immune systems have been damaged, such as those who have HIV/AIDS or are taking immunosuppressive medication, are also at an elevated risk of HEV infection [25]. Similarly, because of the natural deterioration that occurs with advancing age, older people’s immune systems are weaker than those of younger people, which makes them more vulnerable to infections. This puts older people at a greater risk of HEV infection [26]. Similarly, immune systems are already compromised in patients with a liver disease, such as hepatitis B or C, cirrhosis, or liver failure, and they have an increased chance of being infected with HEV. The virus is capable of causing more damage to the liver, which may result in more severe consequences and in some instances even death. When it comes to people suffering from liver problems, an HEV infection might be more severe and last for a longer period of time than it does in healthy people. Additionally, the virus may worsen preexisting liver abnormalities, which can lead to a more rapid development of liver disease as well as an increased risk of liver failure. The acute liver failure that may result from an HEV infection in certain people is a condition that poses a significant risk to their lives [27,28].

Immune response proteins such as SAMSN1, CCLs, and CXCLs play a vital role in signaling leukocyte migration at the infection site [29,30]. Similarly, TNFs and interferon receptors facilitate the initiation of cell pathways associated with infections [31,32,33]. These genes were previously studied concerning HBV, HCV, liver diseases, and viral-induced cancers [34,35,36], but no study showed this gene with expression profiles in HEV infections. The expression of the seven immune response genes CCL2, CCL5, CXCL10, CXCL15, TNF, IFNGR1, and SAMSN1 was evaluated through a real-time PCR. Their expression in patients was compared with 124 healthy individuals which showed the up-regulated expression of all studied genes in HEV-infected individuals. An assessment of the genes’ expression of the viral load further indicated the direct association between the HEV viral load and CCL2, CCL5, CXCL10, CXCL15, and IFNGR, whereas an indirect association was found between SAMSN1 and TNF. The outcome suggested that the expression of these genes was up-regulated in HEV-infected patients. Still, the expression of CCL2, CCL5, CXCL10, CXCL15, and IFNGR increased, and the expression of TNF and SAMSN1 decreased compared to the increased viral load.

In cells, these genes play the role of initiating the immune response to mitigate microbial infection. The cytokine receptor pathway initiates the NF-κB pathway which prompts the expression of TNF and several interleukins. Upon interacting with its receptors, TNF induces signal transduction through the NF-κB and ERK pathways which lead to the expression initiation of CCLs, CXCLs, and other immune response genes that cause the recruitment of immune cells [37]. TNF and INFG also play a role in initiating apoptosis in infected cells [38]. In the present study, the expression of TNF decreased with an increased viral load, suggesting that HEV might initiate a mechanism that supports its survival and HEV infection. Therefore, there is a need to perform more detailed mechanistic studies that help us further understand the underlying signaling of HEV infections. Furthermore, there is a need to perform the analysis on larger cohort. For a more in-depth look at the risk factors, severity, and effects of HEV infection, as well as the impact on host immune response genes, larger studies are required. This will aid in bettering the management of HEV infections and informing the development of appropriate therapies.

The current study also studied the relationship between patient physical parameters and immune response gene expression in HEV patients. The results indicated the elevated expression of all genes in patients between the ages of 20 and 50 and in female patients. These outcomes must be validated on larger cohorts involving more risk factors and disease parameters.

## 5. Conclusions

The present study evaluated the impact of an HEV infection on the expression profile of seven immune response genes: CCL2, CCL5, CXCL10, CXCL15, TNF, IFNGR1, and SAMSN1. The expression of all the genes was up-regulated with females and patients between the age of 20 and 50 at the highest risk. The study outcomes provided valuable mechanistic insight that has the potential to be used for delineating prognostic as well as diagnostic markers for HEV infections. A thorough comprehension of the disease mechanism can also facilitate constructing a novel therapeutic approach for the HEV disease.

## Figures and Tables

**Figure 1 pathogens-12-00392-f001:**
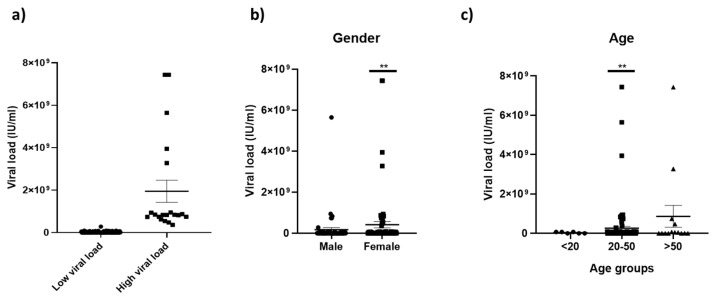
HEV viral load in the Pakistani population (represented by dots, squares and triangles in the figures). (**a**) Patient’s distribution as per low and high viral load where most patients had a viral load below the mean (316822634). (**b**) Association of viral load with gender. (**c**) Association of viral load with patient age. ** indicates the significance below 0.01.

**Figure 2 pathogens-12-00392-f002:**
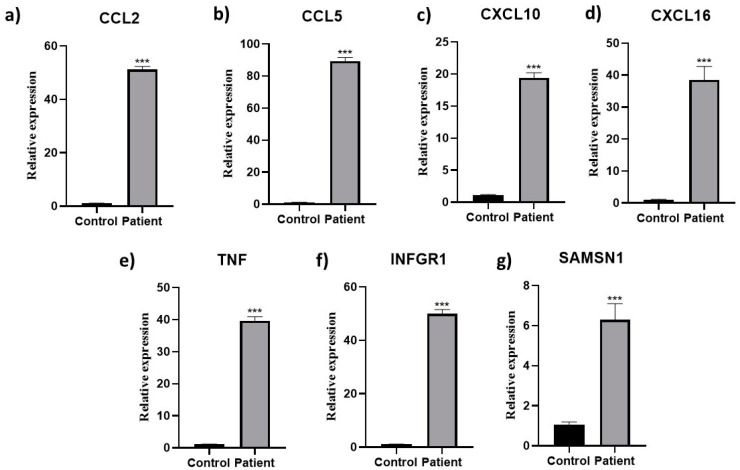
Immune response gene expression in HEV patients. The association of gene expression of (**a**) CCL2, (**b**) CCL5, (**c**) CXCL10, (**d**) CXCL15, (**e**) TNF, (**f**) IFNGR1, and (**g**) SAMSN1 with HEV rel-ative to control. Significance below 0.001 is depicted with ***.

**Figure 3 pathogens-12-00392-f003:**
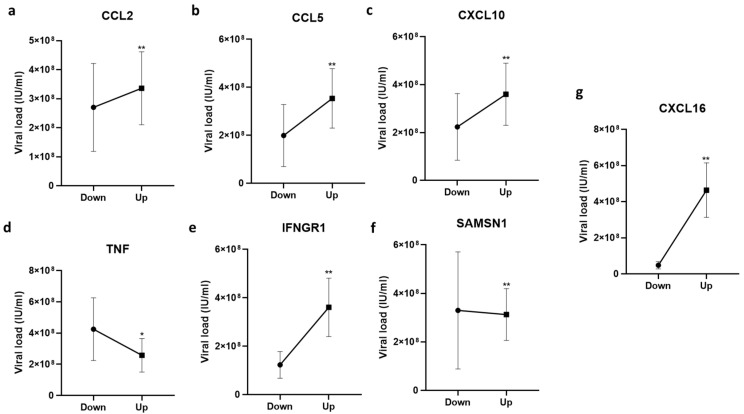
Immune response gene expression and HEV viral load. The viral load is studied in patients with higher expression as well as lower expression of the genes. The association of gene expression of (**a**) CCL2, (**b**) CCL5, (**c**) CXCL10, (**d**) TNF, (**e**) IFNGR1, (**f**) SAMSN1, and (**g**) CXCL15. Ct value above 25 is considered down-regulated, while below 25 is taken as up-regulated. Study group is plotted at x-axis and mean viral loads at y-axis. Significance was calculated by computing student-t-test (patients with up-regulated expression vs down-regulated expression). Significance below 0.01 and 0.001 is depicted with * and **, respectively.

**Figure 4 pathogens-12-00392-f004:**
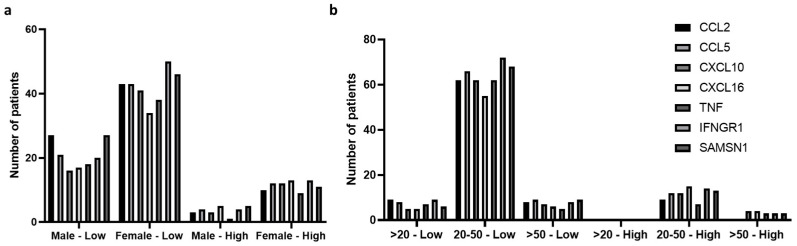
Immune response genes (CCL2, CCL5, CXCL10, CXCL15, TNF, IFNGR1, and SAMSN1) expression as per patients’ physical features. (**a**) Gene expression as per patient gender where expression up-regulation is recorded in females. (**b**) Gene expression as per age where expression up-regulation is observed in the age group 20–50 with low viral load. Only patients having up-regulation of the genes CCL2, CCL5, CXCL10, CXCL15, TNF, IFNGR1, and SAMSN1 were distributed in different groups and compared with each other.

**Figure 5 pathogens-12-00392-f005:**
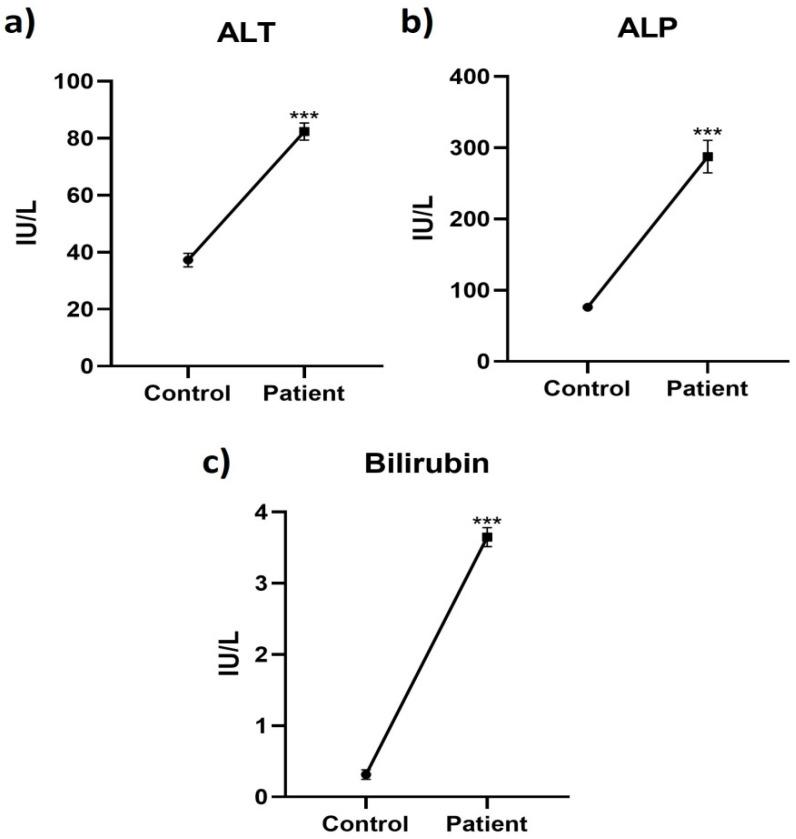
Comparison of the level of markers (**a**) ALT (**b**) ALP and (**c**) Bilrubin between HEV patients and control groups. Symbol “***” represents significance below 0.001.

**Figure 6 pathogens-12-00392-f006:**
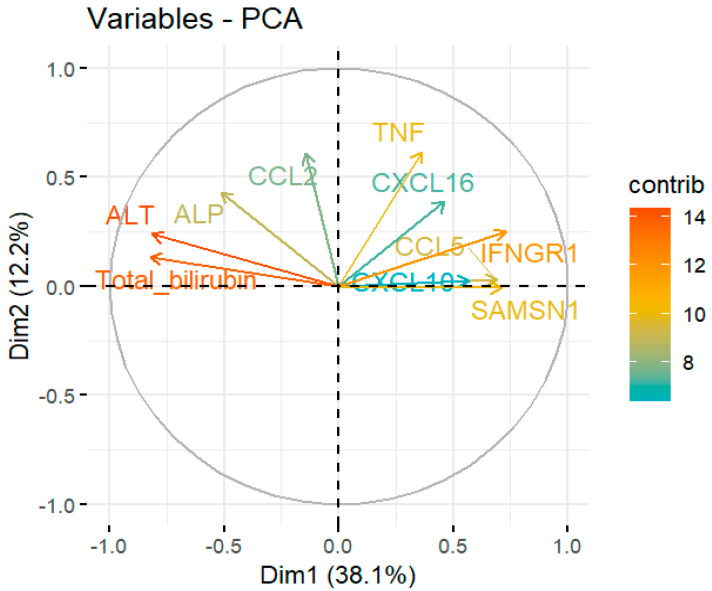
Relationship between liver function and immune response genes (CCL2, CCL5, CXCL10, CXCL15, TNF, IFNGR1, and SAMSN1) expression. Dimensionality analysis indicated the negative relation between liver function markers and immune response genes. Among these genes, TNF, CCL5, IFNGR1, and SAMSN1 have maximum contribution in disrupting liver function.

**Figure 7 pathogens-12-00392-f007:**
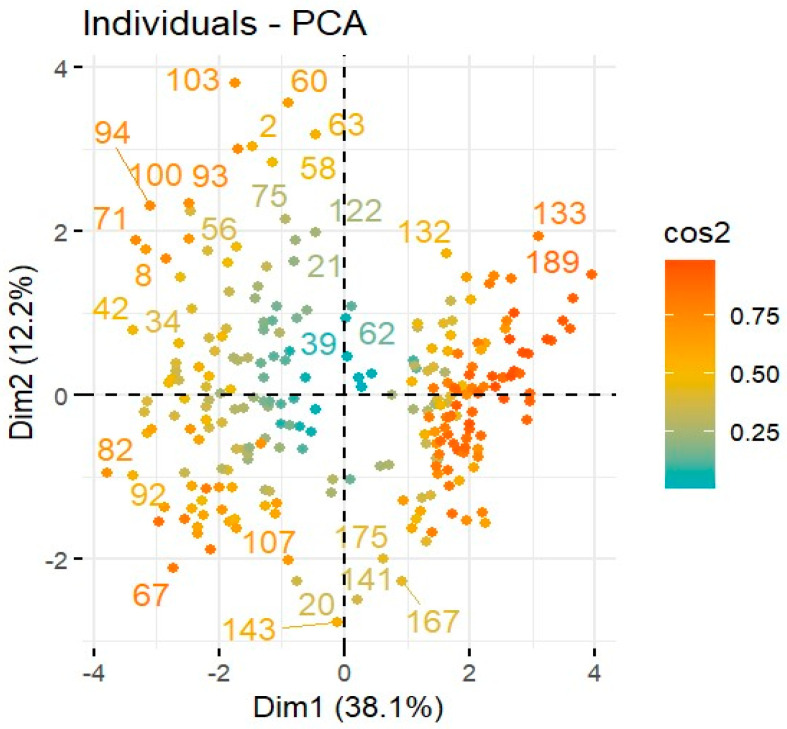
Biplot representation of Principal Analysis Scores (PCA), Dim1 and Dim2 for immune response genes.

**Figure 8 pathogens-12-00392-f008:**
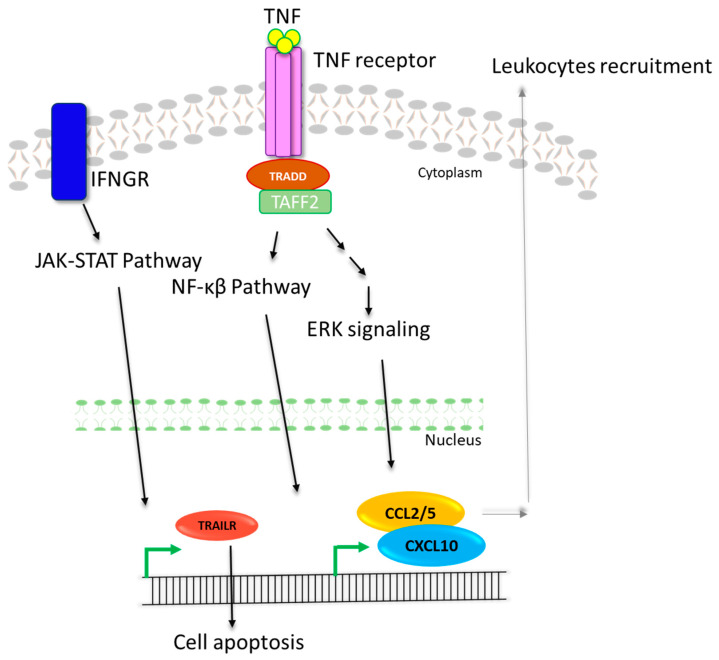
Cellular pathway functioning in response to HEV infection. HEV infection promotes the signaling through TNF and IFN pathways. TNF pathway further activates ERK and NF-κβ pathway and eventually leads to the recruitment of leukocytes. IFN pathway activation causes up-regulation of apoptotic pathway genes leading to cell apoptosis.

**Table 1 pathogens-12-00392-t001:** Information on the participants related to their age and gender.

Group	Age (Mean ± SD)	*p*-Value	Gender
Patients	36 ± 12	<0.001	Males = 56 Females = 74
Control	37 ± 12	<0.0001	Males = 45 Females = 79

**Table 2 pathogens-12-00392-t002:** Expression of immune response gene in HEV patient vs. control.

Group	Expression	Gene Expression in the Number of Patients						
		CCL2	CCL5	CXCL10	CXCL16	TNF	IFNGR1	SAMSN1
Control	Up	17	8	13	3	19	10	11
	Down	107	116	111	121	105	114	113
Patient	Up	92	99	89	84	84	106	99
	Down	38	31	41	46	46	24	31

## Data Availability

The data presented in this study are available on request from the corresponding author.

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
