# Peer review of "Expression Profiles of Hepatic Immune Response Genes in HEV Infection"

_pathogens, 2023, doi:10.3390/pathogens12030392_

Round 1

Reviewer 1 Report

Badshah, Y et al. study the expression of host hepatic genes (CCL2, CCL5, CXCL10, CXCL16, TNF, IFNGR1 and SAMSN1) in patients infected with HEV compare to non-infected patients to understand the progression of the HEV infection and determine biological markers for the disease.  Overall, the findings are interesting. However, some of the results could address in a more profitable way, in order to increase the scientific significance of the manuscript. Hope that the following comments will further increase the quality of their manuscript.

Comments:

1.     The authors can show the qPCR results, with a graph showing the relative expression of each gene studied. They need to indicate the genes used as references.

2.     In figure 2 the authors show that there is a down-regulation of TNF and SAMSN1 associated with a high viral load, but how do these numbers look in comparison with the control patients, at the end can they can say that there is a total up-regulation as indicated in table 2? 

3.     In figure 3 the graphics showed the gene expression by patients, but they didn’t express a real up-regulation of the genes, can these could be expressed in a different way?

4.     The author should try to determine the genotype of HEV in patients to see if there is a correlation between genotypes and the levels of the expression of the genes in the study.

Minors:

1.     The number of control patients is indicated as 119, however,  the total number in table 1 is 124, also this number is indicated in the abstract. 

2.     The viral load numbers could be expressed in scientific mode, it would be helpful in the reading and comparison of the numbers.

3.     In lane 52 change HEC for HEV.

Author Response

Reviewer 1: Badshah, Y et al. study the expression of host hepatic genes (CCL2, CCL5, CXCL10, CXCL16, TNF, IFNGR1 and SAMSN1) in patients infected with HEV compare to non-infected patients to understand the progression of the HEV infection and determine biological markers for the disease.  Overall, the findings are interesting. However, some of the results could address in a more profitable way, in order to increase the scientific significance of the manuscript. Hope that the following comments will further increase the quality of their manuscript.

Response: We thank the reviewer for the positive feedback. We have incorporated all the comments of the reviewer that has helped us in presenting our work more vividly. The response to each comment is provided below:

Comments:

  1. The authors can show the qPCR results, with a graph showing the relative expression of each gene studied. They need to indicate the genes used as references.

Response: We thank the reviewer for the comment. Three replicate analyses were performed to evaluate gene expression of the CCL2, CCL5, CXCL10, CXCL16, TNF, IFNGR1, and SAMSN1, and the amount of target RNA was normalized with respect to the endogenous control (housekeeping) gene GAPDH. We have calculated the fold change and plotted a relative expression graph of these genes in patients relative to the control (Figure 2, Line 103, Page 4).

  1. In figure 2 the authors show that there is a down-regulation of TNF and SAMSN1 associated with a high viral load, but how do these numbers look in comparison with the control patients, at the end can they can say that there is a total up-regulation as indicated in table 2?

Response: The expression of all studied immune response genes CCL2, CCL5, CXCL10, CXCL16, TNF, IFNGR1, and SAMSN1 was up-regulated in patients. However, in association with HEV viral load, the expression of TNF and SAMSN1 was down-regulated in patients with higher viral load. Initially, we only represented the expression of these genes with respect to the patient number in Table 2, which might be confusing to comprehend the actual expression of these genes in HEV patients. Therefore, we have calculated the fold change compared to the control and shown the fold change as bar charts in figure 2. The fold change calculations indicate that the expression of all genes was up-regulated in HEV patients.

  1. In figure 3 the graphics showed the gene expression by patients, but they didn’t express a real up-regulation of the genes, can these could be expressed in a different way?

Response: Figure 3 depicts the comparison of the number of patients as per their different features (such as age and gender) and viral load that has up-regulation of the genes. The overall analysis indicated that the number of patients that has up-regulation of these genes was higher in females with low viral load and in patients belonging age 20-50 with low viral load. However, we have presented a detailed analysis of the expression of these genes in HEV patients in figure 2 in results section 2.2.

  1. The author should try to determine the genotype of HEV in patients to see if there is a correlation between genotypes and the levels of the expression of the genes in the study.

Response: Authors thank the reviewer for the comment. All HEV samples included in the present study belonged to genotype 1a as the number of genotype 1a was most prevalent and the required sample number to perform the study was available. The inclusion of genotype 1a for the study is also indicated in the methodology section.

Minors:

  1. The number of control patients is indicated as 119, however, the total number in table 1 is 124, also this number is indicated in the abstract.

Response: The actual number of control is 124. I have corrected the mistake in the whole manuscript. (Page 1, Line 16; page 2, line 60; page 6, line 170, and page 6, line 201).

  1. The viral load numbers could be expressed in scientific mode, it would be helpful in the reading and comparison of the numbers.

Response: We appreciate the reviewer for thoroughly reading the manuscript. Viral load in figures is presented in scientific mode. We have also converted viral load values presented in the text into scientific mode.

  1. In lane 52 change HEC for HEV.

Response: As indicated, typing error has been corrected.

Reviewer 2 Report

The manuscript entitled ‘Gene expression profiling of HEV infection’ describes an analysis of expression of pro-inflammatory genes in blood isolated from patients with and without HEV infection. Data on correlations between viral loads, patients’ ages, genders etc are reported. Overall high HEV loads are associated with increased expression of the panel of genes. The study could be of interest and utility but requires revision before it is suitable for publication. Specific points are the following:

1                    The origin of the mRNA that is subjected to RT-PCR is stated as being the liver. Although possible, other sources such as T cells from the blood are also possible contributors to the analysed mRNA.

2                    Related to the above point, the methods describe extraction of RNA from whole blood that was frozen after collection. Freeze-thawing is likely to lead to cell lysis and release of mRNA from T cells and other possible sources of the transcripts. Clearer definition of the source of the nucleic acids needs to be provided.

3                    The study carried out analysis on patients who were grouped according to very basic categories (age, gender etc.). It would have been useful for the study to have included clinical features as well as standard markers of hepatitis (e.g. bilirubin, ALT etc.). Understanding how the gene expression markers correlate with clinical features would assist with providing better insights into management of HEV infection. Importantly, HEV infection is associated with high mortality in pregnant women, and it would be valuable if the gene expression analysis could be used to define prognostic markers in these individuals.

4                    The statistical analysis is not clearly presented. P values are provided, but throughout the manuscript it is not clear which groups are being compared.

5                    The title of the manuscript is too broad. The impression is created that the study is a whole genome analysis, which of course it is not.

6                    Minor points include the following:

a.       Exponential notation of numbers, e.g. viral loads, should be provided.

b.      In Figure 1c, for the group of patients on the left, <20 is surely the correct rather than >20.

c.       There are grammar errors that should be corrected.

Author Response

Reviewer 2: The manuscript entitled ‘Gene expression profiling of HEV infection’ describes an analysis of expression of pro-inflammatory genes in blood isolated from patients with and without HEV infection. Data on correlations between viral loads, patients’ ages, genders etc are reported. Overall high HEV loads are associated with increased expression of the panel of genes. The study could be of interest and utility but requires revision before it is suitable for publication. Specific points are the following:

Response: We thank the reviewer for the positive comments. Based on the comments, we have tried to improve the manuscript and have responded to each comment below:

  1. The origin of the mRNA that is subjected to RT-PCR is stated as being the liver. Although possible, other sources such as T cells from the blood are also possible contributors to the analysed mRNA.

Response: Authors appreciate the reviewer for raising a good aspect of the manuscript. The study studied the blood expression of host immune response genes CCL2, CCL5, CXCL10, CXCL16, TNF, IFNGR1, and SAMSN1. The origin of these gene expressions can be the liver or the immune response cells expressed overwhelmingly due to HEV infection. We have corrected the text wherever it implied that the liver is the sole source of these genes’ expression.

  1. Related to the above point, the methods describe extraction of RNA from whole blood that was frozen after collection. Freeze-thawing is likely to lead to cell lysis and release of mRNA from T cells and other possible sources of the transcripts. Clearer definition of the source of the nucleic acids needs to be provided.

Response: For the extraction of RNA, whole blood was used. The purpose was to identify the mRNA transcripts in the blood; so that blood-based expression of genes (CCL2, CCL5, CXCL10, CXCL16, TNF, IFNGR1, and SAMSN1) can be determined. These genes are expressed by hepatic cells as well as innate immune cells that facilitate initiating the necessary immune response against microbial infection. We have skimmed and modified the text wherever it implies that the source of nucleic acid is solely the liver.

  1. The study carried out analysis on patients who were grouped according to very basic categories (age, gender etc.). It would have been useful for the study to have included clinical features as well as standard markers of hepatitis (e.g. bilirubin, ALT etc.). Understanding how the gene expression markers correlate with clinical features would assist with providing better insights into management of HEV infection. Importantly, HEV infection is associated with high mortality in pregnant women, and it would be valuable if the gene expression analysis could be used to define prognostic markers in these individuals.

Response: We are grateful to the reviewer for the comment. The study’s hypothesis was to find the association of immune response genes CCL2, CCL5, CXCL10, CXCL16, TNF, IFNGR1, and SAMSN1 expression and HEV. We further determined the influence of viral load along with age and gender on the expression of these genes in HEV patients. We found up-regulation of all studied genes in HEV patients’ blood. However, the expression of TNF and SAMSN1 was found to be down-regulated with increased viral load. We also found that the expression of these genes was higher in females with low viral load compared to females with high viral load and males. To enhance the clarity of the results, we have included the fold change analysis of these genes in HEV patients in comparison to healthy cohorts (Figure 2). We obtained samples that were positive for HEV, and studying other clinical features, such as pregnancy was not part of the hypothesis; therefore, it was ignored. However, we can design a whole new study to study the association between HEV genotypes and pregnancy.

4 .The statistical analysis is not clearly presented. P values are provided, but throughout the manuscript it is not clear which groups are being compared.

Response: We have improved the text and the figure captions to clearly state the groups being compared and how the analysis was performed. To determine the expression of the genes CCL2, CCL5, CXCL10, CXCL15, TNF, IFNGR1, and SAMSN1 in HEV, the expression of the genes was compared between HEV patients and healthy individuals, and fold change was calculated. ANOVA is applied to calculate significance. To understand the viral load distribution in different groups, the viral load of each patient was plotted through a dot plot, and a student t-test was applied for significance calculation. Similarly, patients were distributed the high and low gene expression group to determine the association between viral load and gene expression. The Ct value means were plotted and compared for significance through a student-t-test. Statistical methodology is revised in the methodology section.

  1. The title of the manuscript is too broad. The impression is created that the study is a whole genome analysis, which of course it is not.

Response: We have revised the title of the manuscript and the new title is “Association of immune response genes’ expression with HEV”.

  1. Minor points include the following:
  2. Exponential notation of numbers, e.g. viral loads, should be provided.

Response:  We appreciate the reviewer for thoroughly reading the manuscript. Viral load in figures is presented in exponential notion. We have also converted viral load values presented in the text into exponential notions.

  1. In Figure 1c, for the group of patients on the left, <20 is surely the correct rather than >20.

Response: The typing error has been corrected in figure 1c.

  1. There are grammar errors that should be corrected..

Response: Whole manuscript has been revised for grammar, syntax errors, and typos.

Round 2

Reviewer 2 Report

Only minor improvement to the manuscript has been achieved by the revisions. Major and minor points of criticism have not been addressed adequately. That is despite the authors indicating that this has been done. Examples are definition of the source of the RNA that was analysed (blood cells vs hepatocytes vs other cells), Exponential notation of numbers has not been changed throughout (e.g. see lines 94-98), groups that are statistically different from each other are not clearly indicated (e.g. Fig. 1 & Fig. 3).

The broad categorisation of the patients limits value of the data. This should have been considered at the planning stages of the study.

Author Response

Only minor improvement to the manuscript has been achieved by the revisions. Major and minor points of criticism have not been addressed adequately. That is despite the authors indicating that this has been done. Examples are the definition of the source of the RNA that was analysed (blood cells vs hepatocytes vs other cells), Exponential notation of numbers that has not been changed throughout (e.g. see lines 94-98), groups that are statistically different from each other are not clearly indicated (e.g. Fig. 1 & Fig. 3).

The broad categorisation of the patients limits value of the data. This should have been considered at the planning stages of the study.

Response:

We greatly apologize for not meeting the expectations of the reviewer. We have again reviewed the previous comments to see what we unintentionally missed. Each comment by the reviewer in the previous round was:

  1. Origin of mRNA

Response: We value the reviewer’s concern regarding the origin of mRNA. We took whole blood and extracted mRNA from it, and mRNA expression of immune response genes in HEV patients’ blood was studied in comparison to healthy individuals.

  1. Source of nucleic acid

Response: We took the whole blood, isolated mRNA from it, and studied the expression of the immune response genes.

  1. Analysis according to other clinical features, such as pregnancy

Response: Reviewer asked us to include more clinical parameters, such as pregnancy, in the study. As data of pregnant ladies were not targeted in this study, con-incidentally pregnant women also did not participate. However, we value the reviewer’s suggestion, so we increased the number of liver markers in the study and performed the analysis to determine their association with immune response genes in HEV patients. We initially evaluated the ALT, ALP, and Bilirubin concentration difference between patients and control (supplementary data 1). We then performed Principal component analysis-based distribution analysis through RStudio to study the relationship between liver function disruption and immune response gene expression in HEV patients.  The relevant methodology is included in the methodology section (page 10, Line 327-331):

Liver function markers (ALT, ALP, and Bilirubin) were compared between healthy individuals and HEV patients through GraphPad and simple-t-test was employed to determine association. Similarly, principal component analysis was performed through RStudio using Procomp command and relationship between immune response genes’ expression and liver dysfunction in HEV was determined.

Results are described in section 2.4, and the graph is illustrated in figure 5.

  1. Statistical analysis

Response: Reviewer asked us to explain the statistical analysis used and which groups were being compared. We initially explained the groups in the figure captions, which could have been a better strategy as it left the reader dubious. Therefore, we have now explained the statistical tests used for the analysis in the methodology section (pages 9 & 10, Lines 275-290):

Statistical analysis was performed via GraphPad Prism version 8.0.1 and Microsoft Excel 2016 (Microsoft cooperation). The expression of genes in patients as well as healthy individuals was calculated through the double-delta method, and fold change was determined. Two-way ANOVA was applied to determine the significance. The mean ± standard deviation (SD) along with ANOVA was performed to find the association of mean viral load in different categories, such as age and gender, and to determine the significant association with age or gender. To perform the analysis, patients were distributed into high and low viral load categories, and each group was evaluated for its association with viral load. Similarly, a simple t-test was performed to find the association of gene expression with viral load. Patients were distributed into low and high gene expression groups and evaluated for their significant association with the viral load. Patients with a viral load above the mean viral load were distributed in high viral load groups. In contrast, patients with a viral load below the mean viral load were distributed into low viral load groups. Ct values above 25 were taken as expression down-regulation, and Ct values below 25 were taken as expression up-regulation. All the graphs were plotted through GraphPad.

  1. Manuscript title

Response: Reviewer asked us to modify the title and make it more specific. We changed the title to “Association of immune response genes’ expression with HEV.”

  1. Exponential notation

Response: Reviewer asked to change the representation style of viral load into exponential notation. Despite our efforts, we apologize that some viral load values remained unchanged. We have rectified our mistake and ensured that all of the viral load values are represented in exponential notion. Modifications made are at:

  • Page 2, Line 63, 67
  • Page 3, Line 94, 95, 96, 98, 99
  1. Figure 1c

Response: Reviewer asked us to correct the figure wrap text as the sign of “lesser than” was wrong. We corrected that as per the reviewer’s advice.

  1. Grammatical error

Response: The reviewer suggested we improve the grammatical errors. We revised the whole manuscript and corrected all the grammar mistakes. Furthermore, we have again revised the manuscript to correct any left grammar errors.

Round 3

Reviewer 2 Report

Statistical analysis remains confusing. The asterisks indicating significance do not clearly indicate which data sets are being compared. Also, based on error bars, differences seem unlikely to be significant. An example is in Figure 3f: viral loads seem similar in the up and down groups but there is a statistical significance indicated. Similarly in Figure 3d, e & g, up and down groups each have statistical differences indicated. What is different to what?

The PCA data is an interesting addition. However the presentation of the data is unconventional, which again complicates interpretation.

Author Response

Reviewer 2:

Comment 1: Statistical analysis remains confusing. The asterisks indicating significance do not clearly indicate which data sets are being compared. Also, based on error bars, differences seem unlikely to be significant. An example is in Figure 3f: viral loads seem similar in the up and down groups but there is a statistical significance indicated. Similarly in Figure 3d, e & g, up and down groups each have statistical differences indicated. What is different to what?

Response: In figure 3, we calculated the significant association of immune response genes’ expression with viral load. Patients were distributed in two groups: one with up-regulated gene expression and the second with down-regulated gene expression. Both groups were compared through t-tests, and the difference between the means of the two groups was calculated, and a t-statistic was derived from this difference. The t-statistic is compared to a critical value from a t-distribution table to determine whether the difference is statistically significant. Suppose the t-statistic is greater than the critical value. In that case, the null hypothesis is rejected, and it is concluded that there is a significant difference between the means of the two groups.

For plotting the graph, the mean viral load of both groups was calculated and plotted at the y-axis and groups at the x-axis. Error bars are indicative of variability in the data. One possible reason for it is the sample size which is also one of the limitations of our study. We have discussed this point in the discussion section. Further, the principle of the t-test is also briefly explained in the Figure 3 caption.

Comment 2: The PCA data is an interesting addition. However, the presentation of the data is unconventional, which again complicates interpretation.

Response: In a Principal Component Analysis (PCA) graph, the dimensions refer to the axes of the scatter plot used to visualize the results of the PCA. The purpose of PCA is to reduce the dimensionality of a dataset by transforming the variables into a set of uncorrelated components that capture the most important information in the data. The analysis integrates the variables’ variability score; therefore, its representation will be complex. Figure 5 is the simplest representation of the analysis.

We, however, value the reviewer’s opinion. Therefore, we have added more descriptions regarding the interpretation of the data. Further, we have included the PCA-individual plot in supplementary data 2. This plot only describes the overall distribution of variables based on the scores its algorithm has generated. This graph also indicated that genes SAMSN1, IFNGR1, CCL5, and TNF are positively correlated.